# How can county-level industrial clusters in China promote urban-rural integration?—A study on the configuration effects based on fsQCA

Gaoyang Liang[ID][1]*, Mingqiang Xing[2], Jianqiang Zhao[1]

1 School of Public Administration, Hebei University of Economics and Business, Shijiazhuang, Hebei, China, 2 School of Management, Xi'an Jiaotong University, Xi'an, Shanxi, China

* lianggaoyang@hueb.edu.cn

## Abstract

This study explores how county-level industrial clustering in China promotes urban-rural integration by identifying four key pathways: industrial linkage, technology transfer, employment coordination, and service balance. Drawing on a dataset of 343 counties from 2013 to 2023 and applying fuzzy-set Qualitative Comparative Analysis (fsQCA), the research uncovers the configuration effects of these mechanisms. The results reveal three primary models: the industry–technology–employment-driven model, the industry–employment-driven model, and the industrial linkage-dominant model. Further, configuration analyses by cluster type indicate that industrial linkage and service balance are central in agriculture-oriented clusters; employment coordination and technology transfer are critical in industry-oriented clusters; and service balance and employment coordination jointly drive integration in service-oriented clusters. Temporal analysis over the past decade demonstrates a steady strengthening of industrial linkage, a rapid increase in technology transfer, persistently high levels of employment coordination, and gradual improvement in service balance. These findings provide new insights into the multi-pathway dynamics of urban-rural integration and inform differentiated policy approaches based on cluster types.

## 1 Introduction

As vital intermediary regions between urban and rural systems, county-level areas in China have witnessed notable advancements in both industrial clustering and integration processes in recent years. On the one hand, county-level industries have become increasingly agglomerated, forming distinct clusters oriented around agriculture, manufacturing, and services [1]. These industrial clusters have contributed to local economic growth by enhancing the efficiency of resource allocation. On the other hand, national policy initiatives have steadily advanced urban-rural integration, particularly in education, healthcare, and transportation, thereby narrowing the developmental and resource disparities between regions [2].

**Data availability statement:** All relevant data are within the paper and its Supporting Information files.

**Funding:** Funded by Science Research, Project of Hebei Education Department:"Study on the Driving Mechanism and Nurturing Pathways of Hebei's County Economy to Promote Urban-Rural Integration" (Project No. BJS2024032). The funders had no role in study design, data collection and analysis, decision to publish, or preparation of the manuscript.

**Competing interests:** Funded by Science Research, Project of Hebei Education Department:"Study on the Driving Mechanism and Nurturing Pathways of Hebei's County Economy to Promote Urban-Rural Integration" (Project No. BJS2024032).

Despite these achievements, urban-rural integration still faces substantial challenges. Key barriers include weak industrial linkages, fragmented labor markets, and unequal access to public services—factors that collectively impede deeper integration. As key nodes in the integration process, counties urgently need to understand how industrial clustering can facilitate the effective coordination of urban and rural resources. Current research primarily adopts static, single-factor analyses, offering limited insight into the dynamic mechanisms by which industrial clustering fosters integration.

Building on a comprehensive literature review, this study identifies four key pathways by which county-level industrial clustering facilitates urban-rural integration in China: industrial linkage, technology transfer, employment coordination, and service balance. Accordingly, this study employs fuzzy-set Qualitative Comparative Analysis (fsQCA) to explore the configurational effects of multiple clustering pathways on urban-rural integration.

## 2 Literature review

### 2.1 The impact of county-level industrial clustering on urban-rural economic coordination

Existing research suggests that county-level industrial clustering fosters urban-rural integration primarily by enhancing regional economic coordination mechanisms. Agricultural clustering contributes to modernization by optimizing value chains and boosting the value-added output of agricultural products [3]. Industrial clustering supports rural industrialization and the relocation of industries, thereby reinforcing urban-rural industrial linkages and enhancing economic efficiency at the county level. Service clustering—propelled by market expansion and the upgrading of service sectors—intensifies economic linkages and promotes integrated urban-rural development [4].

However, much of the existing literature adopts a single-industry lens and overlooks the internal structures and inter-industry synergies within clusters, thereby failing to capture the differentiated contributions of various clustering types to urban-rural integration.

### 2.2 The impact of county-level industrial clustering on urban-rural factor mobility

Industrial clustering at the county level facilitates the integration and efficient utilization of urban and rural resources by optimizing the mobility of labor, capital, and technology. Studies indicate that agricultural clustering helps absorb surplus rural labor into industrial and service sectors, thereby stimulating rural economic growth [5]. In parallel, industrial and service clusters not only serve as labor absorbers but also promote technological diffusion and capital accumulation in rural areas via feedback mechanisms, thus propelling urban-rural integration [6].

However, existing studies largely emphasize the scale of factor mobility, paying insufficient attention to flow quality, matching efficiency, institutional barriers, and path dependence in the mobility process.Consequently, the configurational mechanisms that shape factor flows across urban and rural areas remain underexamined.

## 2.3 The impact of county-level industrial clustering on equalization of urban-rural public services

Service clustering plays a critical role in advancing the equalization of public service provision across urban and rural regions. Research suggests that service-oriented clustering improves access to education, healthcare, and cultural amenities, thereby reducing disparities in public resource allocation. By contrast, agricultural and industrial clusters exert a more limited impact on public service distribution, highlighting the necessity of targeted policy interventions to achieve greater service equity [7].

Nevertheless, much of the literature relies on static indicators such as facility counts or coverage rates, with limited exploration of dynamic mechanisms, institutional collaboration, and the bidirectional relationships between public service provision and industrial clustering.

## 2.4 Summary of the literature

In summary, the existing literature exhibits several limitations:

**Predominance of Single-Path Analyses and Lack of Configurational Thinking: Most studies** examine the impact of industrial clustering on economic coordination, factor mobility, or public services separately, without considering how different pathways interact to jointly promote urban-rural integration.

**Insufficient Mechanism Identification and Lack of a Dynamic Perspective:** Few studies explore how industrial clustering fosters integration through different mechanisms—such as industrial chain collaboration, technology diffusion, labor transfer, and service delivery—under varying conditions, and there is a general lack of investigation into dynamic evolution processes.

**Weak County-Level Focus:** Compared to studies at the city or regional scale, systematic research focusing on counties as intermediary spaces connecting urban and rural areas remains limited. The complex effects arising from the interplay of multiple mechanisms within counties have yet to be fully uncovered.

Therefore, this study focuses on counties as the unit of analysis, exploring how different configurational mechanisms—industrial linkage, technology transfer, employment coordination, and service balance—jointly promote urban-rural integration using the fsQCA approach. This aims to address the current gap in understanding the complex mechanisms underpinning multi-pathway effects.

## 3 Theoretical

Drawing on existing literature and empirical insights, this study proposes that county-level industrial clustering advances urban-rural integration through four key pathways: industrial linkage, technology transfer, employment coordination, and service balance (Fig 1).

### 3.1 Industrial linkage

The industrial linkage pathway emphasizes the coupling and integration of urban and rural areas along the industrial chain. According to growth pole theory, economic activity first concentrates in core regions—such as central cities or county hubs—and then diffuses outward to peripheral areas. County-level industrial clusters frequently serve as regional growth poles, facilitating the "polarization–diffusion" process through vertical integration and industrial chain extension. Leading urban enterprises drive the involvement of surrounding rural resources in the division of labor, facilitating the extension of primary industries (agriculture) into processing and service sectors, thereby realizing industrial linkage. In addition, spatial interaction theory underscores the persistent exchange of resources, information, and labor among proximate geographic areas [8]. Through transportation and information networks, such flows enhance inter-industrial connectivity. As intermediaries between urban and rural areas, counties capitalize on their geographic and institutional

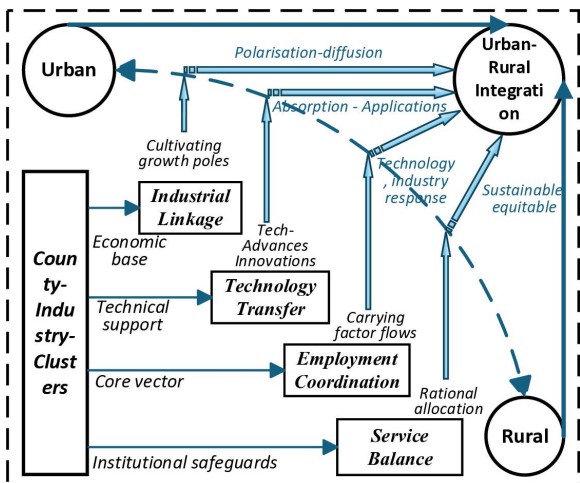

**Fig 1. Theoretical paths of county industrial agglomeration affecting urban-rural integration.**

advantages to integrate factor markets, streamline regional labor divisions, and construct collaborative industrial systems, thereby enabling efficient and complementary resource allocation [9].

### 3.2  Technology transfer

Technology transfer represents the flow of knowledge and innovation critical to urban-rural integration. According to diffusion of innovation theory, new technologies typically originate in urban centers and gradually disseminate through spatial and economic networks. Situated between cities and villages, counties act as key nodes in this diffusion chain. On the one hand, advanced technologies and managerial practices disseminate from cities via county-based enterprises, industrial parks, and collaborative platforms, driving rural modernization and industrial upgrading [10]. On the other hand, county governments and enterprises engage in learning, adaptation, and reinvention, moving from passive acceptance to active application and local adaptation of technologies. This process facilitates two-way knowledge exchange and strengthens rural absorptive capacity, thereby bridging the urban-rural technological divide [11].

### 3.3  Employment coordination

Employment coordination forms the social foundation for urban-rural integration and serves as the vehicle for industrial linkage and technology transfer. Lewis's dual-sector model suggests that economic growth is propelled by the reallocation of labor from traditional agriculture to modern industrial sectors. China's longstanding dualistic urban-rural structure has hindered such transitions, making counties critical channels for labor mobility [12].County-level clusters absorb surplus rural labor, thereby enhancing employment stability and reducing underemployment. Simultaneously, through vocational training, skills development, and labor market construction, counties enhance labor allocation efficiency across urban and rural spaces, while also facilitating the flow of production factors and technologies [13].

### 3.4  Service balance

Service balance underpins urban-rural integration by ensuring equitable access to essential public services, including education, healthcare, cultural amenities, and transportation. According to theories of regional balanced development and spatial equity, urban-rural integration requires the equitable allocation of service resources to address developmental imbalances. Integrated public service provision and coordinated policy efforts help rural populations share the benefits of

urbanization, while also drawing urban capital and technology into rural settings. This reciprocal interaction fosters mutual development and advances the goal of shared prosperity between urban and rural regions [14].

## 4  Methodology

### 4.1  Research method and data sources

Fuzzy-set Qualitative Comparative Analysis (fsQCA) is a case-oriented comparative method grounded in set theory and Boolean algebra. It is particularly well-suited for examining complex causal interactions—such as complementarity, substitution, and suppression—and understanding how these interactions collectively shape the outcome. As such, it serves as an effective analytical tool for identifying joint effects and configurational causality in social science research [15].

This study uses a sample of 343 counties in China. Given the temporal consistency of causal patterns, data from six time points—2013, 2015, 2017, 2019, 2021, and 2023—are selected to capture longitudinal variation. The fsQCA method is applied to examine the relationships. To mitigate the problem of limited diversity—where the number of empirical cases is smaller than the total number of potential condition combinations—it is standard practice to limit the number of antecedent conditions to four to seven in medium-N studies. Accordingly, this study selects four key mechanisms—industrial linkage, technology transfer, employment coordination, and service balance—as antecedent conditions. These variables are used to identify core and peripheral configurations that drive urban-rural integration at the county level.

The primary data sources include the China City Statistical Yearbook, China County Statistical Yearbook, county-level yearbooks, and socioeconomic development bulletins. To address missing data in certain counties or years, linear interpolation and moving average techniques are applied for imputation. These imputed values comprise approximately 6.7% of the dataset and are evenly distributed across regions and years, indicating no systematic clustering. To ensure robustness, additional tests were conducted by excluding counties with high imputation rates and applying alternative interpolation methods. The results demonstrate minimal variation in indicator values and structural composition, with consistent model fit across specifications.

### 4.2  Variable measurement

**4.2.1  Urban-rural integration.**  Drawing on the primary indicators from China's Urban Development Index System [16], this study further designs a set of secondary indicators tailored to measure urban-rural integration, as presented in Table 1.

Population dynamics represent a foundational dimension of urban-rural integration, assessed through the interaction of spatial distribution, total residency, and mobility patterns. Comparative analysis of population density between counties and cities reveals spatial settlement equilibrium, while total resident population metrics reflect demographic scale and regional shifts. Population mobility indicators capture both the intensity and direction of demographic flows across urban and rural areas. When examined collectively, these indicators not only delineate the static spatial configuration of populations but also reveal the dynamic processes underpinning the deepening and expansion of population integration.

Economic integration acts as a key driver of urban-rural convergence, encompassing metrics such as total output, per capita income, fiscal spending, fixed asset investment, and social consumption. Comparing county and city GDP reveals the degree of economic convergence, while per capita income disparities indicate shifts in relative living standards. Fiscal expenditure ratios illuminate the degree of equitable resource allocation by governments, and patterns of fixed asset investment indicate the extent to which infrastructure and industrial systems are harmonized. Moreover, the alignment in social consumption patterns between urban and rural residents further signifies the convergence of economic behaviors. Together, these metrics portray a comprehensive landscape of economic integration, spanning production, distribution, investment, and consumption processes.

**Table 1. Measurement indicator system for urban-rural integration.**

| Overall Goal | Primary Indicator | Secondary Indicator | Measurement Method |
|---|---|---|---|
| Urban-Rural Integration | Population Integration | Population Distribution | County population density/ City population density |
| | | Population Size | County permanent population/ City permanent population |
| | | Population Mobility | County mobile population/ City mobile population (Mobile population = permanent population in current period − permanent population in previous period) |
| | Economic Integration | Economic Scale | County GDP/ City GDP |
| | | Per Capita Income | County per capita income/ City per capita income |
| | | Fiscal Expenditure | County fiscal expenditure/ City fiscal expenditure |
| | | Fixed Asset Investment | County fixed asset investment/ City fixed asset investment |
| | | Social Consumption | County total social consumption/ City total social consumption |
| | Social Integration | Compulsory Education | County student-teacher ratio in compulsory education/ City student-teacher ratio in compulsory education |
| | | Library Holdings | County average library holdings/ City average library holdings |
| | | Medical Health | County hospital bed density/ City hospital bed density |
| | Spatial Integration | Electricity Consumption | County per capita electricity consumption/ City per capita electricity consumption |
| | | Mobile Communication | County mobile phone users/ City mobile phone users |
| | | Internet Access | County broadband access users/ City broadband access users |
| | | Administrative Area | County administrative land area/ City administrative land area |
| | Ecological Integration | Oxygen Demand Emissions | County industrial oxygen demand emissions/ City industrial oxygen demand emissions (per unit GDP) |
| | | Smoke and Dust Emissions | County industrial smoke and dust emissions/ City industrial smoke and dust emissions (per unit GDP) |
| | | Carbon Dioxide Emissions | County industrial carbon dioxide emissions/ City industrial carbon dioxide emissions (per unit GDP) |

Indicators for urban-rural integration are divided into five dimensions: population, economy, society, space, and ecology. Each is measured using the ratio of county-level to city-level data. Data Source: China County Statistical Yearbook (2013–2023).

Social integration, essential for inclusive development, is measured through access to compulsory education, cultural resources, healthcare services, and basic living infrastructure. Variations in teacher-student ratios within compulsory education reveal discrepancies in educational equity, while differences in per capita library holdings illustrate access to cultural resources. Similarly, the availability of hospital beds per capita serves as a proxy for healthcare accessibility, and per capita electricity consumption indirectly reflects living standards and modernization levels. These interconnected indicators collectively provide a multi-dimensional portrait of the progress in social integration across education, culture, health, and everyday life domains.

Spatial integration is driven by both physical and digital infrastructure connectivity, as well as adjustments in administrative governance frameworks. Indicators such as mobile communication usage and broadband internet penetration offer critical insights into the degree of information infrastructure convergence across urban and rural spaces. Furthermore, changes in administrative area configurations reflect institutional adjustments aimed at fostering more cohesive governance frameworks. The synergy between infrastructure interconnectivity and administrative boundary optimization not only mitigates physical and institutional barriers but also facilitates the seamless flow of resources, information, and public services, thereby substantively advancing spatial integration.

Ecological integration reflects the alignment of urban and rural regions in advancing environmentally sustainable development practices. This dimension is evaluated through emissions intensity metrics, including ammonia, smoke, and sulfur dioxide emissions per unit of GDP. The comparative decline in these pollutants' emission levels across urban and

rural areas signals a convergent trajectory toward green industrial transformation and environmental stewardship. Beyond environmental quality improvements, ecological integration reflects a deeper commitment to jointly advancing ecological civilization and sustainable development within the broader framework of urban-rural integration.

To measure urban-rural integration and each primary indicator dimension, this study combines Principal Component Analysis (PCA) and the Entropy Weight Method. First, PCA is employed to reduce the dimensionality of the secondary indicators within each dimension, extracting principal components that represent integration characteristics and enhance explanatory power. Second, the Entropy Weight Method calculates the objective weights of each indicator based on information entropy, reflecting their variability and information contribution. The final indicator weight $W_i$ is determined by the following fusion formula:

$$W_i = \beta W_{i-PCA} + (1-\beta) W_{i-Ent}.$$

Where $W_i$ represents the final weight of the i-th indicator, $W_{i-PCA}$ is the weight derived from PCA, $W_{i-Ent}$ is the weight from the entropy method, and $\beta$ is the combination coefficient (set at 0.5 in this study, giving equal weight to PCA and entropy results).

**4.2.2 County-level industrial clustering.** This study measures the degree of county-level industrial clustering using the Location Quotient (LQ) method (see Table 1), calculating the clustering levels of the primary, secondary, and tertiary industries. Based on the highest LQ value among the three sectors for each county, industrial clustering types are classified into three categories: agriculture-oriented clusters (highest concentration in the primary industry), industry-oriented clusters (highest concentration in the secondary industry), and service-oriented clusters (highest concentration in the tertiary industry) [17].

**4.2.3 Mechanism variables.** To further explore the pathways through which county-level economies influence urban-rural integration, this study selects four mechanism variables, with detailed measurement methods shown in Table 2.

To quantify the interaction intensity across industrial, technological, employment, and service dimensions, the Coupling Degree ($C$) is used as the core metric. The coupling degree is calculated as follows:

$$C = [2 \times (X \times Y)^{(1/2)}]/(X + Y).$$

Where $X$ and $Y$ represent the variable values for urban and rural areas under a specific mechanism dimension, and $C$ indicates the degree of coupling between the two. In social systems research, the coupling degree is widely used to measure the interaction, coordination, and synchronous development between two systems [18].

A value of $C$ approaching 1 indicates a high degree of coordination between urban and rural areas, suggesting similar development trends and potential direct linkages, functional complementarities, or synergistic mechanisms. For instance, a high coupling degree in urban-rural patent indicators may suggest strong bidirectional technological flows or collaborative innovation. Conversely, a $C$ value close to 0 reflects significant developmental disparities between urban and rural variables, indicating restricted factor flows, fragmented mechanisms, or severe lags on one side.

The four mechanism variables are defined as follows:

**(A) Industrial Linkage**

Industrial linkage optimizes the urban-rural industrial chain, promoting resource sharing and complementarity. The degree of industrial linkage reflects the integration and interaction among various industries across urban and rural areas, contributing to joint economic development. Measuring coupling degree in this dimension allows for an accurate assessment of the extent of industrial integration and the joint utilization of resources and production factors [19].

**Table 2. Measurement of county-level industrial clustering and mechanism variables.**

| Variable | Measurement | Explanation |
|---|---|---|
| County Industrial Clustering | County-Level Industrial Cluster Density: The Location Quotient (LQ) is used to measure the concentration level of a particular industry in a specific county. The formula is as follows: $LQ_i = (E_{i,j}/E_j)/(E_{i,T}/E_T)$<br>$LQ_i$: The Location Quotient (i.e., the industrial concentration) of industry i in county j.<br>$E_{i,j}$: The output value of industry i in county j.<br>$E_j$: The total output value of county j.<br>$E_{i,T}$: The output value of industry i nationwide.<br>$E_T$: The total output value of all industries nationwide. | |
| Industrial Linkage | Industrial linkage = AVG (urban-rural agricultural linkage coefficient + urban-rural industrial linkage coefficient + urban-rural service linkage coefficient). | Reflects the degree of industrial integration and the extent of upstream-downstream industrial chain linkages between urban and rural areas. Analyzes the integration, complementarity, and extension among industries. |
| | - Urban-rural agricultural linkage coefficient: Coupling degree between "urban agricultural GDP" and "county agricultural GDP". | |
| | - Urban-rural industrial linkage coefficient: Coupling degree between "urban industrial GDP" and "county industrial GDP". | |
| | - Urban-rural service linkage coefficient: Coupling degree between "urban service GDP" and "county service GDP". | |
| Technology Transfer | Technology transfer = AVG (urban-rural technological achievement acceptance coefficient + urban-rural enterprise vitality coefficient + urban-rural technology transaction coefficient). | Reflects the extent of technological diffusion, innovation-driven capabilities, and enterprise vitality across urban and rural areas. |
| | - Urban-rural technological achievement acceptance coefficient: Coupling degree between "urban effective patent grants" and "county effective patent grants". | |
| | - Urban-rural enterprise vitality coefficient: Coupling degree between "urban new enterprise creation" and "county new enterprise creation". | |
| | - Urban-rural technology transaction coefficient: Coupling degree between "urban technology transaction volume" and "county technology transaction volume". | |
| Employment Coordination | Employment coordination = AVG (urban-rural labor force coordination coefficient + urban-rural income coordination coefficient). | Reflects the spatial matching and distribution pattern of labor force and income between urban and rural areas. |
| | - Urban-rural labor force coordination coefficient: Coupling degree between "urban employed population" and "county employed population". | |
| | - Urban-rural income coordination coefficient: Coupling degree between "urban disposable income" and "county disposable income". | |
| Service Balance | Service balance = AVG (urban-rural fiscal support coordination coefficient + urban-rural education coordination coefficient + urban-rural healthcare coordination coefficient). | Reflects the balance level of public service investment between urban and rural areas, and evaluates disparities in educational and healthcare resources. |
| | - Urban-rural fiscal support coordination coefficient: Coupling degree between "urban public service fiscal expenditure" and "county public service fiscal expenditure". | |
| | - Urban-rural education coordination coefficient: Coupling degree between "urban compulsory education student-teacher ratio" and "county compulsory education student-teacher ratio". | |
| | - Urban-rural healthcare coordination coefficient: Coupling degree between "urban hospital bed density" and "county hospital bed density". | |

This table defines key variables for urban-rural integration, measured using location quotients, coupling coefficients, and ratio indicators to reflect coordination between urban and rural areas.

## (B) Technology Transfer

Technology transfer enables the diffusion and spillover of advanced urban technologies into counties, facilitating enterprise-level technology adoption and enhancing rural labor skills through training programs, thereby improving overall productivity. Measuring the coupling degree of technological innovation adoption helps quantify the speed and effectiveness of technology diffusion from urban to rural areas, revealing the potential for urban-rural technological collaboration and development [20].

## (C) Employment Coordination

Employment coordination expands vocational training and employment opportunities, promoting the bidirectional flow of labor and optimizing employment resource allocation. Measuring coupling degree in this dimension provides insights into the balance of employment resources and income distribution between urban and rural areas, highlighting the degree of labor and income coordination [21].

## (D) Service Balance

Governments enhance service balance by increasing fiscal expenditures, improving infrastructure, and promoting shared platforms for education and healthcare. Service balance aims to facilitate the mutual flow and coordinated development of service resources, improving the quality of life for residents. Measuring the coupling degree in public services allows for a quantitative analysis of the balance in education, healthcare, and fiscal resource allocation between urban and rural areas, revealing disparities and identifying areas for improvement [22].

## 4.3 Measurement and calibration

**4.3.1 fssQCA model formulation and refinement.** This study adopts fuzzy-set Qualitative Comparative Analysis (fsQCA) to examine the configurational effects of county-level industrial clustering mechanisms on urban-rural integration. Unlike linear regression models that estimate average marginal effects, fsQCA focuses on causal complexity, where outcomes result from combinations of conditions rather Liang G, than single independent variables. The analytical foundation of fsQCA is based on three key principles: set theory, Boolean algebra, and asymmetric causality. Model Structure and Variable Encoding: Let $Y \in [0,1]$ denote the fuzzy-set membership score of urban-rural integration for each county. The causal conditions are defined as fuzzy sets: Industrial Linkage(A), Technology Transfer(B), C: Employment Coordination(C), D: Service Balance(D). Each variable $X_i \in [0,1]$ is calibrated using percentiles (75% = full membership, 50% = crossover, 25% = full non-membership), based on the actual value distribution of county-level indicators. The calibration yields a fuzzy membership matrix **X** where each row represents a county-year case and each column a calibrated condition.

The fundamental model is: **Y = f(A,B,C,D)**. Where **f(x)** denotes the configurational function mapping sets of conditions to the outcome. The fsQCA algorithm identifies sufficient configurations **Pk** such that: $\forall i$, **if Pk (i)=1→Y(i)≥τ**, where $\tau$ is a predefined consistency threshold (typically $\tau \geq 0.8$). Each configuration **Pk** is a conjunctive expression over the fuzzy conditions, such as: **P1 = A·B·C, P2 = A·C~D.** Thus, the model function becomes: **Y = P1 + P2 + … + Pn,** Where "**+**" denotes the logical OR (maximum operator in fuzzy logic), and "**·**" denotes logical AND (minimum operator). The negation **~X** is interpreted as **1−X**. Truth Table and Logical Minimization: With four conditions, the model space includes $2^4 = 16$ logically possible configurations. A truth table is constructed, listing all combinations of high/low membership in conditions and whether they are associated with high membership in the outcome **Y**. A consistency score is computed for each row: **Consistency(Pk) = [∑i min(Pk(i),Y(i))]/[∑iPk(i).** Configurations above the threshold are retained. Then, the Quine–McCluskey algorithm is applied to simplify overlapping configurations into a minimized logical expression, identifying core and peripheral causal paths.

To enhance precision and contextual relevance, this study refines the fsQCA modeling process in several ways: (1) Coupling-Based Condition Construction: Each mechanism variable (e.g., industrial linkage, technology transfer) is defined via a coupling degree **C = (XY)^(1/2)/(X+Y)**, where X and Y represent urban and rural subsystem indicators. This reflects the interaction intensity across regions, allowing the fsQCA model to capture not just magnitude but also interconnectivity. (2)Year-Specific Calibration and Cross-Year Pooling: Calibration is done separately for each year to respect temporal heterogeneity, then pooled into a single panel of 343 county-year cases. This enables both year-specific and cross-temporal pattern recognition. (3) Dual-Solution Strategy:

Both intermediate and parsimonious solutions are generated and compared. Intermediate solutions are emphasized for theoretical consistency, while parsimonious solutions are used for robustness checks. This refined fsQCA model not only maps complex causal pathways from industrial clustering to urban-rural integration but also operationalizes interregional mechanisms through coupling metrics, time-specific calibration, and comprehensive sensitivity testing.

In fsQCA, each condition (i.e., the four mechanism variables) and the outcome (urban-rural integration) are treated as sets, with each case assigned a membership score within these sets. The process of assigning membership scores is known as calibration [23].

**4.3.2 Rationality of calibration method and sensitivity analysis.** In Table 3 and 4, the raw data are transformed into fuzzy sets through calibration, a critical step in applying the fsQCA method. Following common practice in existing

**Table 3. Descriptive statistics and calibration thresholds for condition and outcome variables.**

| Year | Condition and Outcome Variables | Mean | Standard Deviation | Full Membership | Crossover Point | Full Non-Membership |
|---|---|---|---|---|---|---|
| 2013 | Urban-Rural Integration | 1.79 | 4.12 | 1.97 | 2.33 | 1.43 |
| | Industrial Linkage | 0.51 | 1.17 | 0.56 | 0.66 | 0.41 |
| | Technology Transfer | 0.42 | 0.97 | 0.46 | 0.55 | 0.34 |
| | Employment Coordination | 0.57 | 1.31 | 0.63 | 0.74 | 0.46 |
| | Service Balance | 0.33 | 0.76 | 0.36 | 0.43 | 0.26 |
| 2015 | Urban-Rural Integration | 2.56 | 5.89 | 2.82 | 3.33 | 2.05 |
| | Industrial Linkage | 0.59 | 1.36 | 0.65 | 0.77 | 0.47 |
| | Technology Transfer | 0.19 | 0.44 | 0.21 | 0.25 | 0.15 |
| | Employment Coordination | 0.56 | 1.29 | 0.62 | 0.73 | 0.45 |
| | Service Balance | 0.36 | 0.83 | 0.40 | 0.47 | 0.29 |
| 2017 | Urban-Rural Integration | 3.47 | 7.98 | 3.82 | 4.51 | 2.78 |
| | Industrial Linkage | 0.65 | 1.50 | 0.72 | 0.85 | 0.52 |
| | Technology Transfer | 0.56 | 1.29 | 0.62 | 0.73 | 0.45 |
| | Employment Coordination | 0.46 | 1.06 | 0.51 | 0.60 | 0.37 |
| | Service Balance | 0.39 | 0.90 | 0.43 | 0.51 | 0.31 |
| 2019 | Urban-Rural Integration | 3.97 | 9.13 | 4.37 | 5.16 | 3.18 |
| | Industrial Linkage | 0.68 | 1.56 | 0.75 | 0.88 | 0.54 |
| | Technology Transfer | 0.61 | 1.40 | 0.67 | 0.79 | 0.49 |
| | Employment Coordination | 0.50 | 1.15 | 0.55 | 0.65 | 0.40 |
| | Service Balance | 0.42 | 0.97 | 0.46 | 0.55 | 0.34 |
| 2021 | Urban-Rural Integration | 4.02 | 9.25 | 4.42 | 5.23 | 3.22 |
| | Industrial Linkage | 0.72 | 1.66 | 0.79 | 0.94 | 0.58 |
| | Technology Transfer | 0.68 | 1.56 | 0.75 | 0.88 | 0.54 |
| | Employment Coordination | 0.54 | 1.24 | 0.59 | 0.70 | 0.43 |
| | Service Balance | 0.48 | 1.10 | 0.53 | 0.62 | 0.38 |
| 2023 | Urban-Rural Integration | 4.36 | 10.03 | 4.80 | 5.67 | 3.49 |
| | Industrial Linkage | 0.76 | 1.75 | 0.84 | 0.99 | 0.61 |
| | Technology Transfer | 0.72 | 1.66 | 0.79 | 0.94 | 0.58 |
| | Employment Coordination | 0.66 | 1.52 | 0.73 | 0.86 | 0.53 |
| | Service Balance | 0.53 | 1.22 | 0.58 | 0.69 | 0.42 |

The data presented in this table are the results calculated using fuzzy-set Qualitative Comparative Analysis (fsQCA) software. The values for Full Membership, Crossover Point, and Full Non-Membership are calibrated based on the direct method of calibration recommended in fsQCA. The variables include both condition and outcome variables across different years (2013–2023), and the calibration thresholds were set according to empirical distributions and theoretical justifications.

**Table 4. Comparison of calibration sensitivity analysis results.**

| Calibration Scheme (Full Membership/ Crossover Point/ Full Non-Membership) | Number of Major Paths | Number of Core Conditions | Consistency | Coverage |
|---|---|---|---|---|
| 75%/ 50%/ 25% (Original Setting) | 4 | 3 | 0.87 | 0.65 |
| 70%/ 50%/ 30% (Sensitivity Test 1) | 4 | 3 | 0.84 | 0.64 |
| 80%/ 50%/ 20% (Sensitivity Test 2) | 4 | 2 | 0.86 | 0.62 |

Under different calibration schemes, the number of major paths and core conditions remains stable or fluctuates only slightly. The consistency and coverage indicators show minor variations, indicating good robustness of the research results. In particular, the original setting (75–50–25) demonstrates the highest overall path coverage, suggesting that this calibration scheme offers relative advantages.

fsQCA literature [24], this study adopts thresholds at the 75th, 50th, and 25th percentiles, corresponding to full membership, the crossover point, and full non-membership, respectively. While this approach has empirical support, further explanation is warranted based on the specific characteristics of the current dataset.

Specifically, selecting the 75th percentile for full membership implies that cases with variable values in the top 25% are considered to fully possess the condition. Conversely, the 25th percentile for full non-membership indicates that cases in the bottom 25% are regarded as lacking the condition. The 50th percentile serves as the crossover point, representing the point of maximum ambiguity. To enhance the rationality of the calibration thresholds, descriptive statistical analysis of the original variable distributions was conducted (see Table 3), confirming that most variables exhibit near-normal or unimodal distributions, thereby validating the chosen percentile thresholds.

Given that fsQCA results can be sensitive to calibration thresholds, a sensitivity analysis was also conducted. Alternative threshold sets of 70%−50%−30% and 80%−50%−20% were applied to recalibrate the data and reanalyze the configurations. As shown in Table 4, the structures and explanatory power (Consistency and Coverage metrics) of the key configuration pathways remained generally stable across different calibration settings, and the identification of core and peripheral conditions was consistent. These results demonstrate the robustness of the chosen calibration thresholds.

**4.3.3 Controlling limited diversity and variable selection.** Limited diversity—a common challenge in fsQCA—arises when the theoretically possible combinations of causal conditions far exceed the number of observed cases [25]. To address this issue, this study carefully selects four antecedent condition variables—industrial linkage, technology transfer, employment coordination, and service balance—to balance theoretical coverage and complexity, thereby mitigating the impact of limited diversity.

The selection of the four antecedent variables follows two principles: (1) theoretical relevance, ensuring that each variable is closely related to different dimensions of urban-rural integration; and (2) operational feasibility and sufficient discriminative capacity, ensuring variability across cases to support effective configurational analysis. To further assess the impact of limited diversity, a comparative analysis was conducted by randomly excluding one variable and examining the resulting three-variable configurations (see Table 5). The analysis revealed that the exclusion of any one variable led to the breakdown of certain key pathways, indicating the synergy and necessity of the four-variable combination for explaining the outcome. Moreover, no dominance of logical remainders in the final solution was observed, suggesting that the variable selection effectively controlled for the limited diversity issue.

## 5 Results and discussion

### 5.1 Analysis of necessary conditions

In fsQCA, a condition is considered necessary if it consistently appears whenever the outcome—urban-rural integration in this case—occurs. Prior to configurational analysis, a necessity test is conducted to ensure that truly necessary conditions are not omitted during the process of logical minimization. Necessity is assessed using consistency scores, with a threshold of 0.9 or above typically required to confirm a condition as necessary.

**Table 5. Comparison of limited diversity control effects.**

| Condition Variable Combination Method | Theoretically Possible Configurations | Actual Sample Coverage Configurations | Number of Logical Remainders | Stable Identification of Core Conditions |
|---|---|---|---|---|
| Four Variables (Industry, Technology, Employment, Service) | 16 | 12 | 1 | Yes |
| Three Variables (Excluding Service Balance) | 8 | 5 | 2 | No |
| Three Variables (Excluding Employment Coordination) | 8 | 6 | 2 | No |

When analyzing with four condition variables, the sample achieves a relatively high coverage rate of theoretical configurations, with a low number of logical remainders, and stably identifies key core conditions. In contrast, the three-variable combinations show lower coverage rates and stability. This further verifies the appropriateness of using four variables to explain the results of urban-rural integration.

As shown in Table 6, none of the four antecedent conditions yielded a consistency score exceeding 0.9 for high-level urban-rural integration, suggesting that no single factor is independently necessary for the outcome. This underscores the importance of conjunctural causation and the need to explore multiple causal configurations. However, for non-high-level urban-rural integration, the consistency score for "~Industrial Linkage" was found to be 0.91, exceeding the 0.9 threshold. This suggests that insufficient industrial linkage is a key constraint hindering urban-rural integration in areas with lower development levels. In such regions, ineffective linkage of urban-rural industrial chains prevents the optimal allocation of resources and labor, thereby obstructing the progress of integration [26].

## 5.2 Analysis of configurational paths

For sufficient conditions, the consistency threshold is typically set at 0.8 or 0.85 to ensure that cases within a given configuration reliably exhibit the outcome. Following standard practice, this study sets the consistency threshold at 0.8 and the PRI (Proportional Reduction in Inconsistency) consistency threshold at 0.7 to reduce contradictory configurations. Regarding case frequency, which should be determined based on the sample size, this study adopts a frequency threshold of 1, given the sample size of 343 counties, which falls within the medium-range case studies.

In fsQCA, three types of solutions can be generated: complex, intermediate, and parsimonious. The parsimonious solution, which includes all logical remainders, may overly simplify the results. The intermediate solution incorporates only those logical remainders consistent with theoretical expectations, offering a more balanced simplification. The complex solution, excluding all logical remainders, tends to yield highly complex results [27].

**Table 6. Necessary condition analysis (Average from 2013 to 2023).**

| Antecedents | High-Level Urban-Rural Integration | | Non-High-Level Urban-Rural Integration | |
|---|---|---|---|---|
| | Consistency | Coverage | Consistency | Coverage |
| Industrial Linkage | 0.85 | 0.85 | 0.27 | 0.28 |
| ~Industrial Linkage | 0.28 | 0.27 | 0.92 | 0.91 |
| Technology Transfer | 0.82 | 0.83 | 0.31 | 0.32 |
| ~Technology Transfer | 0.25 | 0.24 | 0.76 | 0.75 |
| Employment Coordination | 0.79 | 0.80 | 0.26 | 0.27 |
| ~Employment Coordination | 0.31 | 0.30 | 0.85 | 0.84 |
| Service Balance | 0.74 | 0.75 | 0.31 | 0.32 |
| ~Service Balance | 0.35 | 0.34 | 0.70 | 0.69 |

"~" represents the logical relationship "NOT."

The configurational paths leading to high-level and non-high-level urban-rural integration are summarized in Table 7. There are five paths leading to high-level integration and three paths leading to non-high-level integration. The individual and overall consistencies of these paths all exceed 0.9, indicating strong explanatory power for the 343 county cases. The overall coverage of the high-level urban-rural integration configurations is 0.78, meaning that these five pathways explain 78% of the high-integration cases. The overall coverage for non-high-level integration configurations is 0.72, explaining 72% of the corresponding cases.

Intermediate solution paths are denoted by (I), and parsimonious solution paths by (P).

Simplified path codes: P1a (I), P2a (I), P3 (I); P1b (P), P2b (P), P2c (P), P4 (P), P5 (P).

As shown in Table 7, while the intermediate and parsimonious solutions exhibit consistency across several paths (e.g., both P1a and P1b emphasize the core role of industrial linkage in promoting urban-rural integration), there are subtle differences in condition coverage and strength. For instance, in the intermediate solution P2a, "technology transfer" is identified as a core condition, whereas in the parsimonious solution P2b, it becomes a peripheral condition or is absent altogether. This variation may result from the influence of logical remainders on variable selection. Comparative analysis of path consistency and coverage reveals that intermediate solutions offer superior consistency and theoretical explanatory power, whereas parsimonious solutions, despite being more concise, may lose some theoretical logic. Therefore, the subsequent interpretation of paths and conclusions in this study are primarily based on the intermediate solution, with the parsimonious solution serving only as a robustness check.

Further analysis of Table 7 shows that the identified configurations for urban-rural integration possess high explanatory power. The overall coverage for high-level integration paths reaches 0.78, indicating that the identified causal configurations sufficiently explain most high-level integration cases. Similarly, the overall coverage for non-high-level integration paths is 0.72, demonstrating strong discriminatory validity. However, it is noteworthy that some paths exhibit relatively low unique coverage. For example, path P1b has a unique coverage of only 0.03, reflecting significant heterogeneity in urban-rural integration across different counties.

Counties represented by the P1b path may rely on a single industrial advantage to drive integration, such as strong industrial counties in resource-based urban clusters. For these counties, policies should focus on consolidating industrial chain advantages and promoting the spillover effects of industries into rural areas, while also addressing structural weaknesses in technology transfer and service coordination [28].

**Table 7. Configurational solutions for high (and non-high) levels of urban-rural integration.**

| Conditional variable | High-Level Urban-Rural Integration | | | | | Non-High-Level Urban-Rural Integration | | |
|---|---|---|---|---|---|---|---|---|
| | P1a(I) | P1b(P) | P2a(I) | P2b(P) | P2c | P3(I) | P4(P) | P5(P) |
| Industrial Linkage | ● | ● | ● | ● | ● | ⊗ | ⊗ | ⊗ |
| Technology Transfer | ● | • | • | | ⊗ | ⊗ | ⊗ | ⊗ |
| Employment Coordination | ● | ● | • | • | • | ⊗ | ⊗ | ⊗ |
| Service Balance | • | ● | | ⊗ | ⊗ | ⊗ | ⊗ | ⊗ |
| Consistency | 0.99 | 0.96 | 0.93 | 0.91 | 0.92 | 0.98 | 0.96 | 0.93 |
| Raw Coverage | 0.61 | 0.23 | 0.34 | 0.21 | 0.25 | 0.62 | 0.22 | 0.25 |
| Unique Coverage | 0.38 | 0.03 | 0.05 | 0.04 | 0.03 | 0.38 | 0.05 | 0.09 |
| Overall Consistency | 0.95 | | | | | 0.96 | | |
| Overall Coverage | 0.78 | | | | | 0.72 | | |

Significant influencing factors are represented by a large "●". Conditions that only appear in the intermediate solution are identified as marginal conditions, represented by a small "•", indicating secondary influencing factors. The absence of core conditions is represented by a large "⊗", and the absence of marginal conditions is represented by a small "⊗". A "blank" indicates that the condition may or may not appear.

**5.2.1 Analysis of high-level urban-rural integration paths.** Table 7 presents three types of configurational paths:

**(A) "Industry–Technology–Employment" Linked Model (P1a):**

Industrial linkage, technology transfer, and employment coordination jointly serve as core conditions for achieving high-level urban-rural integration, forming a virtuous interaction mechanism among them. Industrial linkage provides the economic foundation, technology transfer enhances the collaborative efficiency of the urban-rural industrial chain through spillover effects, and employment coordination ensures efficient labor flow and matching between urban and rural areas. This path is typical in regions with a solid industrial base, strong technological innovation capacity, and high labor mobility, reflecting a multidimensional association of economy, technology, and employment.

**Case:** *Kunshan City, Jiangsu Province.* Kunshan leverages its advanced manufacturing clusters, such as electronics and information industries, along with accumulated innovation advantages, to drive employment transfers and skills upgrading in surrounding rural areas, thereby establishing an integrated urban-rural development pattern of "urban leading rural, industry-city interaction."

**(B) "Industry–Employment" Linked Model (P1b):**

In this path, industrial linkage serves as the core condition, providing a solid material foundation for urban-rural economic linkages, while employment coordination acts as a supportive condition to enhance labor resource allocation efficiency. It suggests that in regions with relatively complete industrial chains, promoting employment flows and industrial collaboration alone can significantly enhance urban-rural integration, even with weaker technology transfer capacities.

**Case:** *Pidu District, Sichuan Province.* Pidu has established a complete industrial chain based on food processing and modern agriculture, creating stable employment channels between urban and rural areas. Despite limited technological innovation, the development of industrial parks and agricultural enterprises has fostered a positive cycle of "industry promotes employment, employment enhances integration."

**(C) "Industrial Linkage" Dominant Model (P2a, P2b, P2c):**

These paths highlight the critical independent role of industrial linkage in achieving high-level urban-rural integration, while other conditions (such as technology transfer and service balance) play supportive roles. Even with deficiencies in technology transfer or service balance, regions with strong industrial linkage can achieve integration by optimizing industrial chain coordination and deepening urban-rural industrial collaboration.

**Case:** *Yiwu City, Zhejiang Province.* Yiwu relies heavily on its small commodities industrial chain to integrate previously marginal rural areas into the global trade system. Although technological innovation and service balance still require improvement, the city achieves efficient integration and redistribution of resources through a well-established market system and logistics infrastructure.

**Analysis of Non-High-Level Urban-Rural Integration Paths (P3, P4, P5):**

It is notable that "~Industrial Linkage" appears as a core condition in all non-high-level integration paths and was identified as a necessary condition for non-integration outcomes in the necessity analysis. This indicates that the absence of industrial linkage is a major barrier to urban-rural integration. Both P3 and P5 paths lack the combined effect of industrial linkage and technology transfer, with service balance also failing to provide effective support. This reflects severe fragmentation in industrial organization, factor mobility, and value chain integration between urban and rural areas.

**Case:** *Counties in Gansu Province.* Although these areas possess certain agricultural resources, the disconnection between urban and rural industrial systems, lack of effective industrial platforms, and concentration of service facilities in urban centers have seriously hindered the advancement of urban-rural integration.

**5.2.2 Analysis of high-level urban-rural integration paths by county-level industrial clustering types.** Table 8 presents the configurational paths by different types of county-level industrial clustering:

**Table 8. High-level urban-rural integration configurations in different county-level industrial clusters.**

| Condition Variables | Agricultural Cluster-Type | | Industrial Cluster-Type | | Service Cluster-Type | |
|---|---|---|---|---|---|---|
| | Q1a | Q1b | Q2a | Q2b | Q3a | Q3b |
| Industrial Linkage | ● | • | • | • | ● | ● |
| Technology Transfer | • | • | ● | ● | • | ● |
| Employment Coordination | • | ● | • | ● | ● | ● |
| Service Balance | | • | | • | • | • |
| Consistency | 0.91 | 0.94 | 0.94 | 0.97 | 0.96 | 0.98 |
| Raw Coverage | 0.21 | 0.27 | 0.23 | 0.31 | 0.38 | 0.49 |
| Unique Coverage | 0.05 | 0.19 | 0.03 | 0.13 | 0.15 | 0.23 |
| Overall Consistency | 0.94 | | 0.96 | | 0.98 | |
| Overall Coverage | 0.72 | | 0.78 | | 0.79 | |

Same as Table 7.

## (A) Agriculture-Oriented Clustering (Q1a, Q1b):

The agriculture-oriented paths show that industrial linkage and service balance are key conditions for high-level urban-rural integration. In Q1a, service balance acts as a core condition alongside industrial linkage to facilitate efficient resource allocation. In Q1b, industrial linkage remains the core, while service balance serves as a peripheral condition. This indicates that optimizing agricultural industrial chains and balancing public service provision are crucial for integration in agriculture-dominated counties.

**Case:** *Wuchang City, Heilongjiang Province.* Relying on its premium rice brand and integrated agricultural industrial chain, Wuchang has achieved full integration of agricultural production, processing, and marketing, while advancing the downward extension of education and healthcare services into rural areas.

## (B) Industry-Oriented Clustering (Q2a, Q2b):

The industrial-oriented paths emphasize the key roles of employment coordination and technology transfer. In Q2a, both employment coordination and technology transfer are core conditions, while in Q2b, technology transfer remains core and employment coordination becomes peripheral. This suggests that strengthening technology diffusion and optimizing labor allocation are crucial for integration in industry-driven counties.

**Case:** *Shunde District, Foshan City, Guangdong Province.* Shunde, a strong manufacturing base, not only prioritizes technological upgrading but also promotes vocational education and skills training across urban and rural areas, facilitating the upward mobility of rural labor into mid-to-high-end positions along the industrial chain.

## (C) Service-Oriented Clustering (Q3a, Q3b):

The service-oriented paths highlight the dual core roles of service balance and employment coordination. In Q3a, service balance is the core condition, while in Q3b, employment coordination is emphasized as core, with service balance as a supporting factor. This indicates that improving service supply and promoting rational labor mobility are key to achieving high-efficiency integration in service-dominated counties.

**Case:** *Tonglu County, Zhejiang Province.* Based on the development of modern services such as express delivery and e-commerce, Tonglu has enhanced its urban-rural infrastructure, education, and healthcare systems, while promoting rural employment through e-commerce entrepreneurship, forming a service-driven integration pathway.

**5.2.3 Temporal evolution of different configurational paths.** Fig 2 illustrates the frequency changes and growth trends of the four integration pathways between 2013 and 2023:

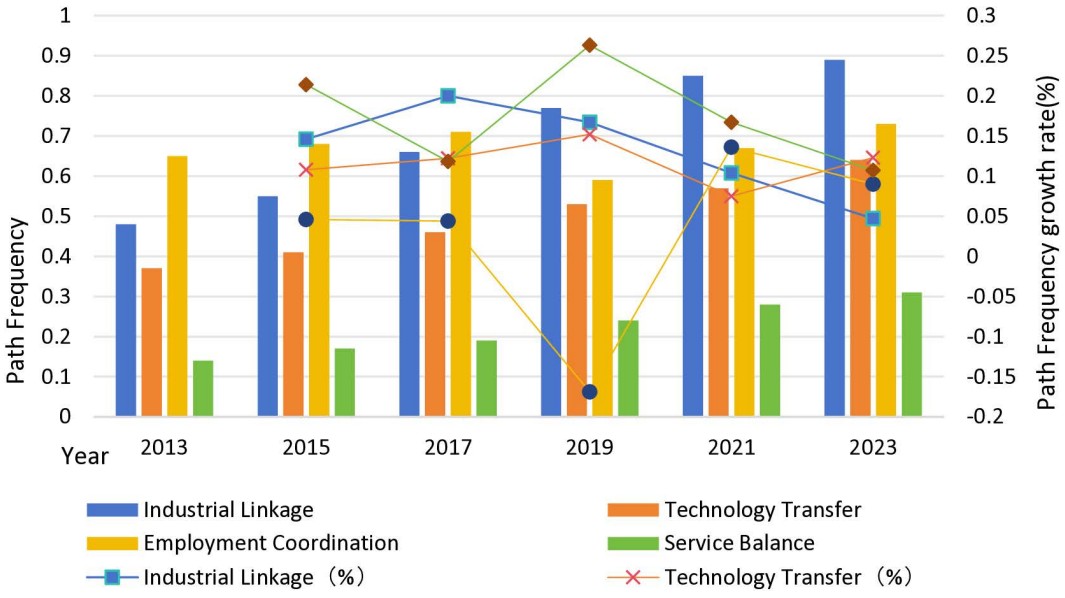

**Fig 2. Frequency of occurrence of each pathway in different years.**

### (A) Industrial Linkage Continues to Strengthen as the Main Axis of Integration:

The frequency of industrial linkage rose from 0.48 in 2013 to 0.89 in 2023, an 85.4% increase over the decade, with a peak growth of 20% between 2015 and 2017.

This trend aligns with China's continuous efforts to integrate urban and rural industries, especially during the 14th Five-Year Plan period, supporting the extension of urban industries into rural areas and the development of shared industrial parks and supply chains. The rise of rural e-commerce and cultural tourism has further facilitated the inflow of urban capital, technology, and brands into rural areas.

### (B) Technology Transfer Steadily Improves, Driven by Policy:

The frequency of technology transfer rose from 0.37 in 2013 to 0.64 in 2023, a 73.0% increase, with a notable surge (+15.2%) between 2017 and 2019.

This is closely related to national policies promoting the "Technology to the Countryside" initiative, shared information infrastructure, and the transformation of agricultural scientific achievements. By 2022, over 90% of counties nationwide had been covered by the "science and technology commissioners" program, significantly enhancing rural areas' capacity for technological adoption.

### (C) Employment Coordination Maintains High Levels with Structural Fluctuations:

The frequency of employment coordination remained high, increasing from 0.65 in 2013 to 0.73 in 2023, a 12.3% rise. Although there was a temporary decline (−16.9%) between 2017 and 2019, the trend later reversed, indicating the persistent importance of labor mobility for integration.

Driven by policies encouraging "return-to-hometown entrepreneurship," labor backflow has gradually strengthened employment coordination within counties, especially in service-oriented counties.

### (D) Service Balance Gradually Improves, with Weak Foundations but Significant Potential:

The frequency of service balance rose from 0.14 in 2013 to 0.31 in 2023, a remarkable 121.4% increase, the fastest among the four mechanisms, with a peak growth rate of 26.3% between 2017 and 2019.

This progress benefits from the continuous expansion of public service access into rural areas, providing "non-physical but equivalent" service supplementation. Nevertheless, service balance remains the weakest mechanism, particularly in agriculture-based and resource-based counties, where further investments in public service infrastructure are urgently needed.

## 6 Conclusion

Using fuzzy-set Qualitative Comparative Analysis (fsQCA), this study identifies and validates four primary pathways through which county-level industrial clustering in China facilitates urban-rural integration: industrial linkage, technology transfer, employment coordination, and service balance. The key conclusions are as follows:

### 6.1 Industrial linkage as the core mechanism for promoting urban-rural integration

Multiple pathways reveal the multidimensional mechanisms driving urban-rural integration. The "Industry–Technology–Employment" pathway highlights the synergistic effects among industrial linkage, technology transfer, and employment coordination, while the "Industry–Employment" pathway emphasizes the precise allocation between industries and labor factors. Even in regions with relatively weaker technology and service conditions, high-level industrial linkage alone can effectively support integration. This finding is consistent with Wang and Zhang (2024a), who argue that inter-organizational cooperation within digital green supply chains promotes technology transfer and innovation, further suggesting that deepening cooperation between urban and rural enterprises in green technology and production coordination is crucial for enhancing integration [29].

### 6.2 Service-oriented counties and green development integration

Although the service balance pathway remains relatively weak overall, it shows significant improvement in service-oriented counties, indicating a need to increase public service infrastructure investments in agriculture- and industry-dominated regions. Echoing the "green entrepreneurship" perspective proposed by Wang and Zhang (2024c), future research should focus on the role of Generative Artificial Intelligence (GAI) in knowledge management and green innovation within county-level industrial clusters, driving urban-rural integration toward a "green, efficient, and intelligent" transformation [30].

### 6.3 Place-based strategies for integration

The emphasis in integration pathways varies across agriculture-, industry-, and service-oriented counties. Agriculture-oriented counties rely more on the synergy between industrial linkage and service balance to enhance the efficiency and sustainability of the agricultural value chain. Industry-oriented counties emphasize the mutual reinforcement between technology transfer and employment coordination. Service-oriented counties prioritize service balance and employment coordination to promote the equalization of public service resources. This finding aligns with the growing importance of the "Social" and "Governance" dimensions within ESG frameworks. Some scholars, such as Wang and Zhang (2024b), have explored the role of Generative Artificial Intelligence (GAI) in optimizing ESG performance. Future research should focus on integrating GAI into county-level governance to enhance policy precision and resource allocation efficiency [31].

While this study reveals the configurational effects of multiple pathways through which county-level industrial clustering promotes urban-rural integration using fsQCA, it should be noted that fsQCA's sensitivity to variable selection and condition settings may affect the stability and generalizability of the configurations. Future research could incorporate dynamic panel data and multi-method comparative approaches to deepen the exploration of mechanisms, particularly focusing on the dynamic impact of institutional and policy factors on the process of county-level industrial clustering promoting urban-rural integration.

## Supporting information

**S1 File. The empirical analysis is based on data from the "S1 China County Statistical Yearbook".**
(XLSX)

## Author contributions

**Conceptualization:** Mingqiang Xing.

**Data curation:** Gaoyang Liang, Mingqiang Xing.

**Methodology:** Mingqiang Xing.

**Resources:** Jianqiang Zhao.

**Software:** Gaoyang Liang.

**Supervision:** Jianqiang Zhao.

**Writing – review & editing:** Gaoyang Liang.

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
