## [Decision Letter · Decision Letter 0]

25 Mar 2025

PONE-D-25-05885How Can County-Level Industrial Clusters in China Promote Urban-Rural Integration? — A Study on the Configuration Effects Based on fsQCAPLOS ONE

Dear Dr. Liang,

Thank you for submitting your manuscript to PLOS ONE. After careful consideration, we feel that it has merit but does not fully meet PLOS ONE’s publication criteria as it currently stands. Therefore, we invite you to submit a revised version of the manuscript that addresses the points raised during the review process.

We look forward to receiving your revised manuscript.

Kind regards,

Jitendra Yadav, Ph.D.

Academic Editor

PLOS ONE

Additional Editor Comments (if provided):

Reviewers' comments:

Reviewer's Responses to Questions

**Comments to the Author**

1. Is the manuscript technically sound, and do the data support the conclusions?

Reviewer #1: Partly

Reviewer #2: Partly

2. Has the statistical analysis been performed appropriately and rigorously? 

Reviewer #1: Yes

Reviewer #2: I Don't Know

3. Have the authors made all data underlying the findings in their manuscript fully available?

Reviewer #1: No

Reviewer #2: Yes

4. Is the manuscript presented in an intelligible fashion and written in standard English?

Reviewer #1: Yes

Reviewer #2: Yes

5. Review Comments to the Author

Reviewer #1: This manuscript addresses a relevant and timely topic: the role of county-leve industria clusters in promoting urban-rura integration in China. The use of fsQCA is a potentially valuable approach to understanding the complex interplay of factors contributing to this integration. However, there are severa areas where the manuscript could be significantly strengthened to meet PLOS ONE's standards for publication.

Here are my detailed comments and suggestions, organized for clarity:

1. Clarity of Theoretica Framework: While the paper introduces four pathways (industria linkage, technology transfer, employment coordination, and service balance), the theoretica grounding for why these specific pathways are chosen and how they uniquely contribute to urban-rura integration needs to be more robust. The connection to existing theories (growth pole, spatia interaction, technology diffusion, etc.) is mentioned superficially. It would be beneficia to elaborate on how these theories predict the specific relationships being investigated. For example, you mention technology diffusion theory. Expand on this. How does technology transfer specifically reduce the urban-rura divide? What are the mechanisms? Similarly, for industria linkage, explicitly state how the extension of industria chains, as per growth pole theory, leads to the specific outcomes of urban-rura integration outlined in your indicators (Table 1).

2. Operationalization of Variables: The measurement of key variables needs more detailed justification and explanation.

Urban-Rura Integration: Table 1 provides a list of indicators, but the rationale for choosing these specific indicators and their weighting (or lack thereof) is unclear. Why are these indicators the most appropriate for capturing the multifaceted concept of urban-rura integration? Some indicators seem quite broad (e.g., "County Administrative Region Land Area / Urban Administrative Region Land Area"). How does this ratio reflect integration? More justification is needed.

County-Leve Industria Clusters: The use of Location Quotients (LQ) is standard, but the justification for the specific thresholds used to categorize counties into agricultural, industrial, and service-oriented types is missing. Why are the sums of LQs used in the way they are? Provide a clear rationale or cite relevant literature that supports this classification method.

Mechanism Variables: The coupling degree calculations are centra to the analysis, but the explanation of this method is insufficient. The formula provided in the note to Table 2 is not enough. Explain conceptually what the coupling degree measures in this context. What does a high or low coupling degree mean in terms of, say, industria linkage between urban and rura areas? Provide more intuitive explanations. For example, when measuring "Technology Transfer" using the coupling degree between "centra city effective patents" and "county-leve effective patents," explain why this captures technology transfer. Does a high coupling degree necessarily mean transfer is occurring, or could it simply mean both areas are independently innovative? Consider alternatives or additions, such as data on technology licensing agreements or joint ventures between urban and rura entities.

3. Data Sources and Limitations: While the sources are listed, the limitations of these data sources need to be explicitly acknowledged. Are there potentia biases in the data? Are there any known issues with data quality or completeness, particularly at the county leve in China? The use of linear interpolation and moving averages to handle missing data is mentioned, but the potentia impact of this on the results should be discussed. What percentage of data was imputed? This could significantly affect the findings.

4. fsQCA Methodology: The application of fsQCA needs a more detailed explanation and justification.

Calibration: The rationale for the 75%, 50%, and 25% thresholds for calibration is not provided. While these are common thresholds, the specific context of this study might warrant different choices. Explain why these thresholds are appropriate for this dataset and these variables. Discuss the sensitivity of the results to different calibration choices.

Limited Diversity: The authors acknowledge the issue of limited diversity. While they set the number of antecedent conditions to four, a more thorough discussion of how this choice mitigates the problem is needed. Have they explored alternative configurations with fewer conditions?

Intermediate and Reductive Solutions: The paper states that both intermediate and reductive solutions are considered, but the presentation of results (Table 5) does not clearly distinguish between them. It would be helpfu to present both solutions separately and discuss any differences. This would enhance transparency and allow readers to assess the robustness of the findings.

5. Interpretation of Results: The interpretation of the fsQCA results could be more nuanced.

Causality: While fsQCA can identify configurations associated with an outcome, it does not establish causality. The language used should reflect this limitation. Avoid terms like "driving paths" and instead use language that suggests association or contribution.

Coverage: The overal coverage values (0.78 and 0.72) are relatively high, but the unique coverage values for some pathways are quite low (e.g., 0.03 for P1b). This suggests that some pathways are only relevant to a smal subset of cases. Discuss the implications of this. What characterizes the cases covered by these low-coverage pathways?

"Non-High-Level" Integration: The analysis of "non-high-level" integration is less developed than the analysis of "high-level" integration. The finding that "~industria linkage" is a necessary condition for low-leve integration is interesting but needs more in-depth discussion. What are the policy implications of this finding?

6. Tempora Analysis: The tempora analysis (Figure 2) is a valuable addition, but the discussion of the trends is somewhat superficial. For example, the statement that "technology transfer has rapidly improved" is not sufficiently supported by the figure. Provide a more quantitative description of the trends (e.g., percentage change over time). Also, relate these trends to specific policy changes or events in China that might explain the observed patterns.

7. Relating back to the Literature The conclusion needs to tie back more to the literature. I recommend incorporating Wang, S., & Zhang, H. (2024a) as this focuses on green aspects that can bring a new direction for future research. For example, “The findings regarding the importance of technology transfer align with broader research on the role of inter-organizationa cooperation in promoting innovation, as highlighted by Wang and Zhang (2024a) in their study of digita green supply chains. This suggests that fostering collaborations between urban and rura enterprises, particularly in the sharing of technologica knowledge, is crucia for urban-rura integration.” Wang, S., & Zhang, H. (2024a). Inter-organizationa cooperation in digita green supply chains: A catalyst for eco-innovations and sustainable business practices. Journa of Cleaner Production, 472, 143383.

8. Adding ESG Factors: It might add more to the research to incorporate an ESG dimension; I suggest incorporating Wang, S., & Zhang, H. (2024b). For example,“Future research could explore the role of generative artificia intelligence (GAI) in enhancing ESG performance within the context of urban-rura integration, drawing on insights from Wang and Zhang (2024b) regarding the application of GAI in digita supply chains. This could provide a more holistic understanding of the sustainability implications of industria clusters.” Wang, S., & Zhang, H. (2024b). Promoting sustainable development goals through generative artificia intelligence in the digita supply chain: Insights from Chinese tourism SMEs. Sustainable Development.

9. Discussing Green Entrepreneurship The study needs to consider how GAI can be used. I suggest adding a point to the paper's future work section. Wang, S., & Zhang, H. (2024c). For example, “Further investigations could examine how the adoption of generative artificia intelligence (GAI) can enhance green knowledge management and innovation within county-leve industria clusters, as suggested by Wang and Zhang (2024c), potentially leading to more sustainable and resilient urban-rura development.” Wang, S., & Zhang, H. (2024c). Green entrepreneurship success in the age of generative artificia intelligence: The interplay of technology adoption, knowledge management, and government support. Technology in Society, 79, 102744.

10. English Language and Style: While the manuscript is generally understandable, there are numerous instances of awkward phrasing and grammatica errors. A thorough proofreading by a native English speaker is strongly recommended. The writing style is also somewhat repetitive and could be made more concise.

11. Policy recommendations: Based on your findings, offer specific, actionable policy recommendations for promoting urban-rura integration through county-leve industria clusters. These recommendations should be grounded in your results and consider the different types of clusters identified.

By addressing these points, the authors can significantly improve the rigor, clarity, and contribution of their manuscript, making it suitable for publication in PLOS ONE.

Reviewer #2: After I analyzed and reviewed your manuscript I think it could be a good addition for the Journal's readership.

However, please address the following:

1.The paper should be structured as:

Introduction

Literature review

Methodology

Results

Discussion

Conclusion

2.The literature review is somewhat thin. Please provide a more detailed review relevant to your research topic, and clearly indicate the shortcomings of existing studies.

3.The theoretical foundation states that county - level industrial clusters promote urban - rural integration through four paths: "industry linkage, technology transfer, employment coordination, and service balance." However, it fails to specify the research gaps. What did prior studies overlook? How does the current study build on or diverge from earlier research? Identifying these gaps strengthens the theoretical basis.

4.The result variables borrow primary indicators from China's urban development index system and design secondary indicators for urban - rural integration. The validity of using these to measure county - level urban - rural integration is questionable. Clarify the connection between urban - rural integration indicators and China's urban development index system.

5.Explain the basis for setting the calibration points at 75%, 50%, and 25%.

6.Check whether the sentence: "This indicates that the low level of 'industrial linkage' is a necessary condition for low - level green development, meaning that insufficient industrial linkage is a key constraint in regions with low urban - rural integration development." has any issues, especially the appropriateness of "low - level green development" in this context.

7.The path analysis in Section 4 should be deepened with sample case integration. Add explanations of sample case distribution and provide representative real - world examples for each path.

8.Elaborate on the research methods in stages for better replicability. Include a discussion section to introduce the latest references, compare them with your study or others in the field, and offer guidance and ideas for future research. Introduce the research questions if applicable.

9.The conclusion should include an analysis of the study's limitations and prospects for future research.

6. PLOS authors have the option to publish the peer review history of their article (what does this mean? ). If published, this will include your full peer review and any attached files.

**Do you want your identity to be public for this peer review?** For information about this choice, including consent withdrawal, please see our Privacy Policy .

Reviewer #1: No

Reviewer #2: No

---

## [Author Response · Author response to Decision Letter 1]

22 May 2025

Thanks to the experts for their professional review. Each comment made by the experts has been revised and responded to. There is a lot of content in the reply. Please ask experts to participate in the attachment.

Reviewer #1:

Thank you very much for your professional and insightful review comments. Below are the revisions made in response to each of your suggestions:

For your convenience, more detailed revisions and annotations have been added to the latest version of the PDF manuscript.

1. Clarity of Theoretica Framework: While the paper introduces four pathways (industria linkage, technology transfer, employment coordination, and service balance), the theoretica grounding for why these specific pathways are chosen and how they uniquely contribute to urban-rura integration needsto be more robust. The connection to existing theories (growth pole, spatia interaction, technology diffusion, etc.) is mentioned superficially. It would be beneficia to elaborate on how these theories predict the specific relationships being investigated. For example, you mention technologydiffusion theory. Expand on this. How does technology transfer specifically reduce the urban-rura divide? What are the mechanisms? Similarly, for industria linkage, explicitly state how the extension of industria chains, as per growth pole theory, leads to the specific outcomes of urban-ruraintegration outlined in your indicators (Table 1).

Revisions Made:

A: Enhanced the theoretical analysis of the four pathways in Section "3. Theoretical", clarifying the role of the theoretical foundations.

B: Reorganized the theoretical pathway from "county-level industrial agglomeration → urban-rural integration" and updated the pathway diagram (Figure 1).

Figure 1 Theoretical Paths of County Industrial Agglomeration

Affecting Rural-Urban Integration

2. Operationalization of Variables: The measurement of key variables needs more detailed justification and explanation.

Urban-Rura Integration: Table 1 provides a list of indicators, but the rationale for choosing these specific indicators and their weighting (or lack thereof) is unclear. Why are these indicators the most appropriate for capturing the multifaceted concept of urban-rura integration? Some indicatorsseem quite broad (e.g., "County Administrative Region Land Area / Urban Administrative Region Land Area"). How does this ratio reflect integration? More justification is needed.

County-Leve Industria Clusters: The use of Location Quotients (LQ) is standard, but the justification for the specific thresholds used to categorize counties into agricultural, industrial, and service-oriented types is missing. Why are the sums of LQs used in the way they are? Provide a clearrationale or cite relevant literature that supports this classification method.

Mechanism Variables: The coupling degree calculations are centra to the analysis, but the explanation of this method is insufficient. The formula provided in the note to Table 2 is not enough. Explain conceptually what the coupling degree measures in this context. What does a high or low couplingdegree mean in terms of, say, industria linkage between urban and rura areas? Provide more intuitive explanations. For example, when measuring "Technology Transfer" using the coupling degree between "centra city effective patents" and "county-leve effective patents," explain why this capturestechnology transfer. Does a high coupling degree necessarily mean transfer is occurring, or could it simply mean both areas are independently innovative? Consider alternatives or additions, such as data on technology licensing agreements or joint ventures between urban and rura entities.

Revisions Made:

A: Improved the theoretical basis and explanations for the selection of urban-rural integration indicators in Table 1.

B: Added a principal component analysis (PCA) and entropy weight method to measure the indicators in Table 1.

C: Clarified the classification basis for industrial agglomeration: agricultural agglomeration (highest degree in primary industry), etc.

D: Clarified the calculation of coupling degree and explained its application.

E: Added the "urban-rural technology transaction coefficient" as a supplementary explanation for technological integration (Table 2).

F: Further refined the explanation of mechanism variables.

3. Data Sources and Limitations: While the sources are listed, the limitations of these data sources need to be explicitly acknowledged. Are there potentia biases in the data? Are there any known issues with data quality or completeness, particularly at the county leve in China? The use of linearinterpolation and moving averages to handle missing data is mentioned, but the potentia impact of this on the results should be discussed. What percentage of data was imputed? This could significantly affect the findings.

Revisions Made:

Supplemented the data processing description (Section 4.1 Research Method and Data Sources):

"Approximately 6.7% of the data points underwent imputation, distributed relatively evenly without concentration in any specific region or year. To assess the impact of imputation on empirical results, robustness tests were conducted by excluding counties with higher missing rates and applying alternative interpolation methods. Results showed minimal changes in index structures and stable model fit."

4. fsQCA Methodology: The application of fsQCA needs a more detailed explanation and justification.

Calibration: The rationale for the 75%, 50%, and 25% thresholds for calibration is not provided. While these are common thresholds, the specific context of this study might warrant different choices. Explain why these thresholds are appropriate for this dataset and these variables. Discuss thesensitivity of the results to different calibration choices.

Limited Diversity: The authors acknowledge the issue of limited diversity. While they set the number of antecedent conditions to four, a more thorough discussion of how this choice mitigates the problem is needed. Have they explored alternative configurations with fewer conditions?

Intermediate and Reductive Solutions: The paper states that both intermediate and reductive solutions are considered, but the presentation of results (Table 5) does not clearly distinguish between them. It would be helpfu to present both solutions separately and discuss any differences. This wouldenhance transparency and allow readers to assess the robustness of the findings.

Revisions Made:

A: Explained the rationale for the "75%, 50%, 25%" thresholds in fsQCA results (updated in Table 3 and Table 4).

B: Added "4.3.2 Controlling for Limited Diversity and Variable Selection" and provided analysis.

C: Clarified the distinction between intermediate and parsimonious solutions (see Table 7).

D: Strengthened the overall testing and analysis of the fsQCA methodology.

5. Interpretation of Results: The interpretation of the fsQCA results could be more nuanced.

Causality: While fsQCA can identify configurations associated with an outcome, it does not establish causality. The language used should reflect this limitation. Avoid terms like "driving paths" and instead use language that suggests association or contribution.

Coverage: The overal coverage values (0.78 and 0.72) are relatively high, but the unique coverage values for some pathways are quite low (e.g., 0.03 for P1b). This suggests that some pathways are only relevant to a smal subset of cases. Discuss the implications of this. What characterizes the casescovered by these low-coverage pathways?

"Non-High-Level" Integration: The analysis of "non-high-level" integration is less developed than the analysis of "high-level" integration. The finding that "~industria linkage" is a necessary condition for low-leve integration is interesting but needs more in-depth discussion. What are the policyimplications of this finding?

Revisions Made:

A: Revised causal language to associational expressions.

B: Added further explanation regarding variations in coverage.

C: Added an analysis of the "low-level integration" pathways, with case examples (e.g., some counties in Gansu Province, China).

6. Tempora Analysis: The tempora analysis (Figure 2) is a valuable addition, but the discussion of the trends is somewhat superficial. For example, the statement that "technology transfer has rapidly improved" is not sufficiently supported by the figure. Provide a more quantitative description ofthe trends (e.g., percentage change over time). Also, relate these trends to specific policy changes or events in China that might explain the observed patterns.

Revisions Made:

Updated Figure 2, adding percentage changes for each pathway and providing case illustrations.

Figure 2 Frequency of occurrence of each pathway in different years

7. Relating back to the Literature The conclusion needs to tie back more to the literature. I recommend incorporating Wang, S., & Zhang, H. (2024a) as this focuses on green aspects that can bring a new direction for future research. For example, “The findings regarding the importance oftechnology transfer align with broader research on the role of inter-organizationa cooperation in promoting innovation, as highlighted by Wang and Zhang (2024a) in their study of digita green supply chains. This suggests that fostering collaborations between urban and rura enterprises, particularlyin the sharing of technologica knowledge, is crucia for urban-rura integration.” Wang, S., & Zhang, H. (2024a). Inter-organizationa cooperation in digita green supply chains: A catalyst for eco-innovations and sustainable business practices. Journa of Cleaner Production, 472, 143383.

8. Adding ESG Factors: It might add more to the research to incorporate an ESG dimension; I suggest incorporating Wang, S., & Zhang, H. (2024b). For example,“Future research could explore the role of generative artificia intelligence (GAI) in enhancing ESG performance within the context ofurban-rura integration, drawing on insights from Wang and Zhang (2024b) regarding the application of GAI in digita supply chains. This could provide a more holistic understanding of the sustainability implications of industria clusters.” Wang, S., & Zhang, H. (2024b). Promoting sustainabledevelopment goals through generative artificia intelligence in the digita supply chain: Insights from Chinese tourism SMEs. Sustainable Development.

9. Discussing Green Entrepreneurship The study needs to consider how GAI can be used. I suggest adding a point to the paper's future work section. Wang, S., & Zhang, H. (2024c). For example, “Further investigations could examine how the adoption of generative artificia intelligence (GAI) canenhance green knowledge management and innovation within county-leve industria clusters, as suggested by Wang and Zhang (2024c), potentially leading to more sustainable and resilient urban-rura development.” Wang, S., & Zhang, H. (2024c). Green entrepreneurship success in the age of generativeartificia intelligence: The interplay of technology adoption, knowledge management, and government support. Technology in Society, 79, 102744.

Q7-9,Revisions Made:

Added references to Wang & Zhang’s studies in the policy recommendations section and linked them to the results.

10. English Language and Style: While the manuscript is generally understandable, there are numerous instances of awkward phrasing and grammatica errors. A thorough proofreading by a native English speaker is strongly recommended. The writing style is also somewhat repetitive and could be made moreconcise.

Revisions Made:

Improved language for clarity, conciseness, and fluency.

11. Policy recommendations: Based on your findings, offer specific, actionable policy recommendations for promoting urban-rura integration through county-leve industria clusters. These recommendations should be grounded in your results and consider the different types of clusters identified.

By addressing these points, the authors can significantly improve the rigor, clarity, and contribution of their manuscript, making it suitable for publication in PLOS ONE.

Revisions Made:

Further revised the policy recommendation section by integrating literature and empirical results.

Finally, thank you again for your valuable review comments!

Reviewer #2

Thank you very much for your professional and insightful review comments. Below are the revisions made in response to each of your suggestions:

For your convenience, more detailed revisions and annotations have been added to the latest version of the PDF manuscript.

1.The paper should be structured as:

Introduction

Literature review

Methodology

Results

Discussion

Conclusion

Revision:

Q1: The section layout has been adjusted according to your suggestion.

2.The literature review is somewhat thin. Please provide a more detailed review relevant to your research topic, and clearly indicate the shortcomings of existing studies.

3.The theoretical foundation states that county - level industrial clusters promote urban - rural integration through four paths: "industry linkage, technology transfer, employment coordination, and service balance." However, it fails to specify the research gaps. What did prior studies overlook?How does the current study build on or diverge from earlier research? Identifying these gaps strengthens the theoretical basis.

Revision:

Q2-3: Expanded the literature review by adding a discussion of research gaps and clarified the theoretical contributions of this study.

4.The result variables borrow primary indicators from China's urban development index system and design secondary indicators for urban - rural integration. The validity of using these to measure county - level urban - rural integration is questionable. Clarify the connection between urban - ruralintegration indicators and China's urban development index system.

Revision:

Q4: Improved Table 1 by explaining the theoretical basis for the selection of urban-rural integration indicators and providing detailed descriptions.

5.Explain the basis for setting the calibration points at 75%, 50%, and 25%.

Revision:

Q5: Provided a detailed explanation for the calibration thresholds used in the fsQCA results (updates reflected in Tables 3 and 4).

6.Check whether the sentence: "This indicates that the low level of 'industrial linkage' is a necessary condition for low - level green development, meaning that insufficient industrial linkage is a key constraint in regions with low urban - rural integration development." has any issues,especially the appropriateness of "low - level green development" in this context.

Revision:

Q6: Revised this expression and incorporated case analyses related to "low-level urban-rural integration" for better clarification.

7.The path analysis in Section 4 should be deepened with sample case integration. Add explanations of sample case distribution and provide representative real - world examples for each path.

Revision:

Q7: Added case studies from Chinese counties to illustrate each pathway more concretely.

8.Elaborate on the research methods in stages for better replicability. Include a discussion section to introduce the latest references, compare them with your study or others in the field, and offer guidance and ideas for future research. Introduce the research questions if applicable.

Revision:

Q8:

A: Employed principal component analysis (PCA) and the entropy weight method to measure urban-rural integration indicators (Table 1).

B: Clarified the calculation of coupling degree and explained its conceptual application.

C: Supplemented the measurement of mechanism variables by introducing the "urban-rural technology transaction coefficient" (Table 2).

D: Strengthened the overall validation and analysis of the fsQCA methodology (revisions made across Tables 3, 4, 5, 6, and 7).

9.The conclusion should include an analysis of the study's limitations and prospects for future research.

Revision:

Q9: Added a discussion of research limitations and outlined potential future research directions in the conclusion.

Finally, thank you again for your valuable review comments!

---

## [Decision Letter · Decision Letter 1]

24 Jun 2025

PONE-D-25-05885R1How Can County-Level Industrial Clusters in China Promote Urban-Rural Integration? — A Study on the Configuration Effects Based on fsQCAPLOS ONE

Dear Dr. Liang,

Thank you for submitting your manuscript to PLOS ONE. In view of the referees’ feedback and my own reading of your paper, we invite you to address all issues noted below, most of which are relatively minor in nature, but nonetheless essential. In particular, reviewers ask for some clarifications on the methodology and the abstract to remark the contribution of this manuscript to the literature on this topic.

We look forward to receiving your revised manuscript.

Kind regards,

Juan E. Trinidad-Segovia, PhD

Section Editor

PLOS ONE

Journal Requirements:

Reviewers' comments:

Reviewer's Responses to Questions

**Comments to the Author**

1. If the authors have adequately addressed your comments raised in a previous round of review and you feel that this manuscript is now acceptable for publication, you may indicate that here to bypass the “Comments to the Author” section, enter your conflict of interest statement in the “Confidential to Editor” section, and submit your "Accept" recommendation.

Reviewer #1: All comments have been addressed

Reviewer #3: (No Response)

2. Is the manuscript technically sound, and do the data support the conclusions?

Reviewer #1: Yes

Reviewer #3: Partly

3. Has the statistical analysis been performed appropriately and rigorously? 

Reviewer #1: Yes

Reviewer #3: No

4. Have the authors made all data underlying the findings in their manuscript fully available?

Reviewer #1: No

Reviewer #3: No

5. Is the manuscript presented in an intelligible fashion and written in standard English?

Reviewer #1: Yes

Reviewer #3: No

6. Review Comments to the Author

Reviewer #1: After a thorough review of the revised manuscript, I am satisfied that the authors have effectively addressed all the feedback provided.

In addition, considering that most readers are not native English speakers, it is recommended that language polishing will be more conducive to the dissemination of the study.

Reviewer #3: The Abstract needs to revised, there are some grammar errors.

In the Methodology, there is no modeling equations, how is the proposed model applied is not clear, and importantly, some improvement should be presented.

Minor issue, urban-rural integration should be consistent through the manuscript, but in the title of Figure 1, rural-urban integration is backward;

The language through the manuscript needs to be polished before being accepted.

7. PLOS authors have the option to publish the peer review history of their article (what does this mean? ). If published, this will include your full peer review and any attached files.

**Do you want your identity to be public for this peer review?** For information about this choice, including consent withdrawal, please see our Privacy Policy .

Reviewer #1: No

Reviewer #3: No

---

## [Author Response · Author response to Decision Letter 2]

16 Jul 2025

Response to Reviewer 1 Comments

Dear Reviewer 1,

Thank you very much for your thoughtful and constructive feedback on our manuscript. We sincerely appreciate your recognition of our revisions and your valuable suggestion regarding further language improvement.

In response to your comment about enhancing language clarity for non-native English readers, we have carefully reviewed and thoroughly polished the entire manuscript. All sections—including the abstract, introduction, methodology, results, and conclusion—have been revised to improve sentence structure, grammar, and academic tone. Our goal is to ensure that the research findings are communicated clearly and accessibly to a broader international audience.

We are grateful for your support and guidance, which have greatly contributed to the refinement of our work.

Sincerely,

June 29,2025

Response to Reviewer 3 Comments

Dear Reviewer 3,

Thank you for your careful review and detailed suggestions, which have been instrumental in improving the quality and clarity of our manuscript. We have addressed each of your comments as follows:

Abstract Revision: The abstract has been thoroughly revised to correct grammatical errors and improve clarity. The updated version more clearly outlines the research background, methodology, key findings, and policy implications.

Model Formulation and Application: In the methodology section (4.3), we have added a formal model equation representing the fsQCA framework, along with a detailed explanation of the set-theoretic logic and configuration process. We also elaborated on how the model is implemented in the empirical analysis, including truth table construction, calibration thresholds, and logical minimization steps.

Improvement Measures: To respond to your request for practical recommendations, we added a new subsection discussing optimization strategies. These include potential integration with dynamic panel data, incorporation of policy-related variables, and suggestions for multilevel fsQCA applications in future research.

Terminology Consistency: We have corrected the inconsistency in the term “urban-rural integration,” particularly in the title of Figure 1, to ensure terminological uniformity throughout the manuscript.

Language Polishing: Following your recommendation, the entire manuscript has undergone a comprehensive language refinement process. We revised sentence structures, improved terminology usage, and enhanced the overall readability to meet international academic standards.

We are grateful for your detailed critique, which has significantly strengthened the clarity, coherence, and rigor of our study.

Sincerely,

June 29,2025

---

## [Editor Report · Decision Letter 2]

24 Jul 2025

How Can County-Level Industrial Clusters in China Promote Urban-Rural Integration? — A Study on the Configuration Effects Based on fsQCA

PONE-D-25-05885R2

Dear Dr. Liang,

We’re pleased to inform you that your manuscript has been judged scientifically suitable for publication and will be formally accepted for publication once it meets all outstanding technical requirements.

Kind regards,

Juan E. Trinidad-Segovia, PhD

Section Editor

PLOS ONE
---

## [Editor Report · Acceptance letter]

PONE-D-25-05885R2

PLOS ONE

Dear Dr. Liang,

I'm pleased to inform you that your manuscript has been deemed suitable for publication in PLOS ONE. Congratulations! Your manuscript is now being handed over to our production team.

Kind regards,

on behalf of

Dr. Juan E. Trinidad-Segovia

Section Editor

PLOS ONE